# Effects of an invasive predator cascade to plants via mutualism disruption

Haldre S. Rogers[1,2,3], Eric R. Buhle[4], Janneke HilleRisLambers[2], Evan C. Fricke[1,2], Ross H. Miller[5] & Joshua J. Tewksbury[6,7,8]

Invasive vertebrate predators are directly responsible for the extinction or decline of many vertebrate species, but their indirect impacts often go unmeasured, potentially leading to an underestimation of their full impact. When invasives extirpate functionally important mutualists, dependent species are likely to be affected as well. Here, we show that the invasive brown treesnake, directly responsible for the extirpation of forest birds from the island of Guam, is also indirectly responsible for a severe decline in plant recruitment as a result of disrupting the fruit-frugivore mutualism. To assess the impact of frugivore loss on plants, we compare seed dispersal and recruitment of two fleshy-fruited tree species on Guam and three nearby islands with intact disperser communities. We conservatively estimate that the loss of frugivorous birds caused by the brown treesnake may have caused a 61–92% decline in seedling recruitment. This case study highlights the potential for predator invasions to cause indirect, pervasive and easily overlooked interaction cascades.

[1] Department of Ecology, Evolution, and Organismal Biology, 251 Bessey Hall, Iowa State University, Ames, Iowa 50011, USA. [2] Department of Biology, University of Washington, Seattle, Washington 98195, USA. [3] Department of BioSciences, Rice University, Houston, Texas 77005, USA. [4] Northwest Fisheries Science Center, National Oceanic and Atmospheric Administration, Seattle, Washington 98112, USA. [5] Western Pacific Tropical Research Center, University of Guam, Mangilao, Guam 96923, USA. [6] Colorado Global Hub, Future Earth, Colorado 80309, USA. [7] Sustainability Innovation Lab at Colorado, University of Colorado, Boulder, Colorado 80309, USA. [8] School of Global Environmental Sustainability, Colorado State University, Fort Collins, Colorado 80523, USA. Correspondence and requests for materials should be addressed to H.S.R. (email: haldre@iastate.edu).

Despite invasive species being considered a major threat to biodiversity[1], there is a growing call for the acceptance of invasive species as a component of novel ecosystems in today's changing world[2,3]. This is fuelled, in part, by a debate about the magnitude of their impact[2,3]. This debate largely ignores the fact that we have rarely assessed the full impacts of invaders on native species. Many of the largest impacts are likely to be buried in indirect effects, which often go unstudied due to their diffuse, cryptic nature and the difficulty in isolating and quantifying their magnitude[4,5].

The brown treesnake (*Boiga irregularis*) is a textbook example used to demonstrate the consequences of species invasions. The unintentional introduction of the generalist predator to the Western Pacific Island of Guam (Fig. 1) in the mid-1940s (ref. 6) caused the complete loss of ten of the 12 native forest bird species, and functional extirpation of the remaining two species[7,8], producing a 'silent forest' (Supplementary Table 1). However, nearly all research attention has gone to the study of direct effects, leaving the indirect impacts virtually unstudied (but see refs 9,10). The extirpation of birds from Guam's forests has resulted in the complete loss of seed dispersal services, making it the only place in the world where all native frugivores have been lost from the forest without replacement by non-native species. Given that ∼70% of tree species on Guam have fleshy fruits adapted for dispersal by birds, the impacts of this mutualism disruption could be extensive.

Fruit handling and seed dispersal are the two key ways in which frugivores influence recruitment. First, while consuming seeds, frugivores may increase germination by removing germination inhibitors or scarifying seeds during gut passage[11]. Second, by moving seeds, frugivores shape the spatial distribution of individuals in forests, which influences the species interactions that those individuals experience, including interactions with predators and pathogens concentrated near parent trees[12–15]. Because lower plant survival near conspecifics and at high conspecific density is a pervasive phenomenon in temperate and tropical forests[14,16–18], reductions in dispersal caused by frugivore loss may reduce recruitment[19–21].

We quantified the indirect influence of the brown treesnake on seedling recruitment by considering how birds handle fruit and move seeds, using a combination of manipulative field experiments, nursery trials and comparative observational studies on bird-free Guam and three nearby islands with birds (Saipan, Tinian and Rota, Fig. 1). We focused on two representative native, fleshy-fruited, small-seeded (<8 mm diameter) tree species: *Psychotria mariana* and *Premna serratifolia,* hereafter referred to by their genus. These species were selected because they are moderately common on all four islands and produce adequate fruit crops for experiments. In over 100 h of fruiting tree observations and observations of foraging Mariana crows[22], we observed all five avian frugivore species formerly found on Guam visiting these two tree species on Saipan and Rota (Fig. 1). Seedlings and saplings of both species are rare on Guam relative to nearby islands, consistent with the prediction that disperser loss reduces plant recruitment. However, the abundance of introduced ungulate herbivores is also highest on Guam, which makes it is difficult to attribute recruitment failure solely to mutualism disruption. Therefore, we designed our study to isolate the role of dispersers.

We found that fruits on Guam fall untouched by frugivores, which strongly reduces germination. In addition, seed rain is clustered underneath parent tree canopies, which increases the proportion of seeds experiencing distance-dependent seedling mortality. Combining these field-based estimates of bird loss impacts, we predict a large decline in seedling recruitment. Changes in dispersal patterns shown by these two tree species when their dispersers were lost are likely to reflect patterns found across many other tree species on the island, because the life history traits of our focal tree species (generalist and small-seeded) matches that of the majority of dominant tree species in these forest. This study demonstrates that invasive predators can have large indirect impacts caused by their direct impacts on mutualistic species.

## Results

**Effects of bird ingestion on germination.** Birds ingested one-quarter to three-quarters of all seeds collected in seed

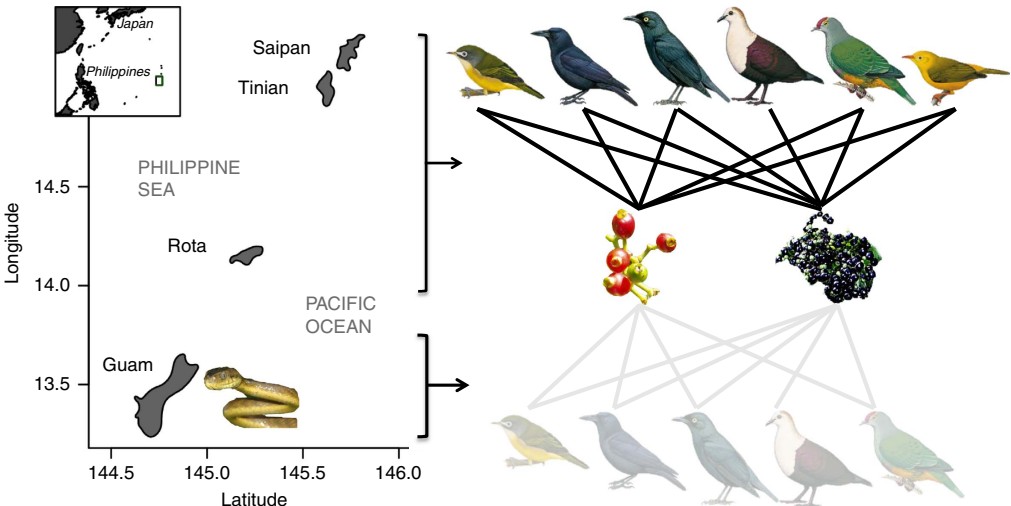

**Figure 1 | Study site.** Guam, the southernmost island in the Mariana Islands, is home to the invasive brown treesnake and thus virtually all forested lands are bird-free, whereas the nearby islands of Saipan, Tinian and Rota are snake-free, and have relatively healthy bird communities. On islands with birds, the primary frugivores (from left to right) for two tree species whose fruits are depicted in the middle, *Psychotria mariana* (left) and *Premna serratifolia* (right), include the bridled white-eye (*Zosterops conspicillatus*), and Rota bridled white-eye (*Zosterops rotensis*; both Zosterops species represented by the bridled white eye in the figure), Mariana crow (*Corvus kubaryi*), Micronesian starling (*Aplonis opaca*), white-throated ground dove (*Gallicolumba xanthonura*; *Premna* only), Mariana fruit dove (*Ptilinopus roseicapilla*) and golden white-eye (*Cleptornis marchei*). As a result of the snake, *Psychotria mariana* and *Premna serratifolia* on Guam have functionally lost all of their seed dispersing partners (note that the Rota bridled white-eye and the golden white-eye were not on Guam before the snake introduction). Latitude and longitude are in degrees north and east, respectively. Snake photograph and bird illustrations[54–59] used with permission.

traps on islands with birds (Fig. 2a,b, Supplementary Table 2), and bird ingestion had large impacts on germination in both tree species (Fig. 2c,d). In a nursery experiment, we compared germination rates of seeds collected from seed traps placed underneath fruiting trees on Saipan, Tinian and Rota. Seeds were separated into three treatments: whole fruits with flesh intact, seeds whose flesh was manually removed, and field-collected seeds without fruit pulp (likely ingested by birds). The probability of germination was two to four times higher for the ingested seeds of *Psychotria* and *Premna* than for whole fruits of each species (Fig. 2c,d, Supplementary Table 3). The effects of bird ingestion extend beyond the simple removal of germination inhibitors present in fruit pulp. Manual de-pulping of the fruit did not produce germination rates similar to those of ingested seeds, potentially due to an increase in seed permeability after scarification[23] or incomplete removal of germination inhibitors. The benefits of germination via ingestion accrue in a spatially inconsistent manner, as the proportion of seeds that have been ingested increases with increasing distance from a conspecific adult (Fig. 2e,f, Supplementary Table 4).

**Seed dispersal kernels.** Birds also affect the spatial pattern of seed rain. We quantified seed dispersal distances by recording seed densities in seed trap arrays (Supplementary Fig. 1) at four forest sites on Guam and three forest sites each on Saipan, Tinian and Rota, and mapping all conspecific trees within 20 m of each trap. Sites were at least 500 m apart. We modelled dispersal by fitting

dispersal kernels to the seed trap data using a hierarchical Bayesian framework[24,25] (Supplementary Table 5,6). Seed rain declined sharply beyond the parent canopy on Guam, whereas seeds were more broadly dispersed on islands with birds (Fig. 3a,b). The mean dispersal distance away from the parent on Guam and the other three islands, respectively, was 0.73 versus 5.37 m for *Psychotria* and 1.18 versus 8.19 m for *Premna* (Supplementary Table 7). These differing dispersal distances imply that 94% of *Psychotria* seeds and 96% of *Premna* seeds on Guam land beneath the canopy of a typical-sized parent, compared with 26% and 40% for these species on islands with birds (Supplementary Table 7). Thus, when frugivorous birds are absent due to the brown treesnake, high seed densities accumulate under adults and a much smaller proportion of the forest floor receives seed rain (Fig. 3c–f). In addition, because the proportion of ingested seeds increases with distance from the parent tree (Fig. 2e,f, Supplementary Table 4), the increased germination conferred by bird ingestion provides an added benefit to seeds at greater distances.

**Distance-dependent mortality.** Because dispersal can increase recruitment by allowing seedlings to escape the detrimental effects of natural enemies such as fungal pathogens, insect herbivores, or mammalian predators associated with parent trees, we examined the consequences of reduced seed dispersal on recruitment. Specifically, we planted seedlings of both tree species in plots near and far from conspecific adults on Saipan, Tinian, Rota, and Guam. Both *Psychotria* and *Premna* seedlings had higher survival in plots far from conspecific trees relative to plots

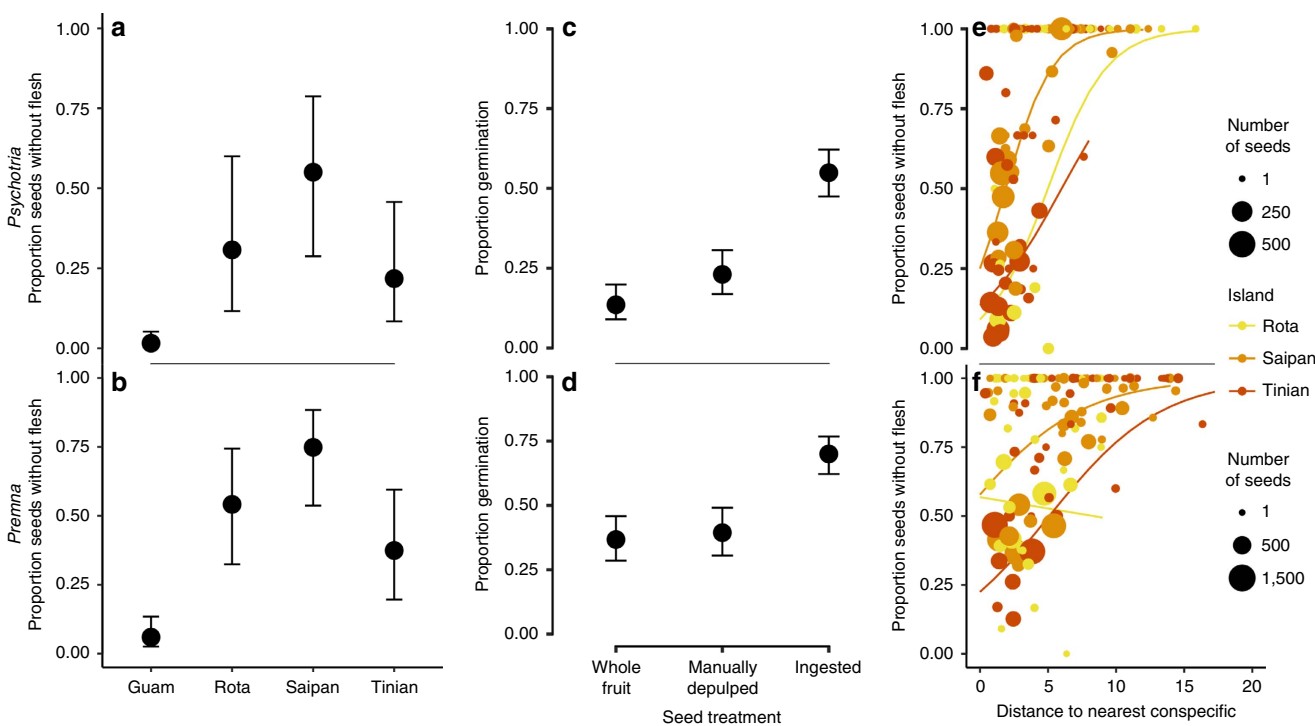

**Figure 2 | Fruit ingestion by birds.** Proportion of seeds in seed traps lacking fruit flesh on Guam (no birds) and on Saipan, Tinian and Rota (with birds), as predicted from generalized linear mixed effects models, for *Psychotria* (**a**) and *Premna* (**b**). The difference between the proportion of seeds without flesh on Guam and that on islands with birds is attributable to bird consumption. *Psychotria* (**c**) and *Premna* (**d**) seeds that have been ingested by birds are more likely to germinate than those that have not, as predicted by generalized linear mixed effects models, and manual flesh removal does not have the same impact as seed ingestion. The proportion of seeds in traps lacking fruit flesh on Rota (yellow), Saipan (light orange) and Tinian (dark orange) is related to the distance to the nearest conspecific for both *Psychotria* (**e**) and *Premna* (**f**). Each point represents a single seed trap, and point size indicates the total number of seeds found in that trap. Model predictions by island are indicated by the lines, limited to the range of distances observed on each island. All error bars represent 95% confidence intervals around model predictions.

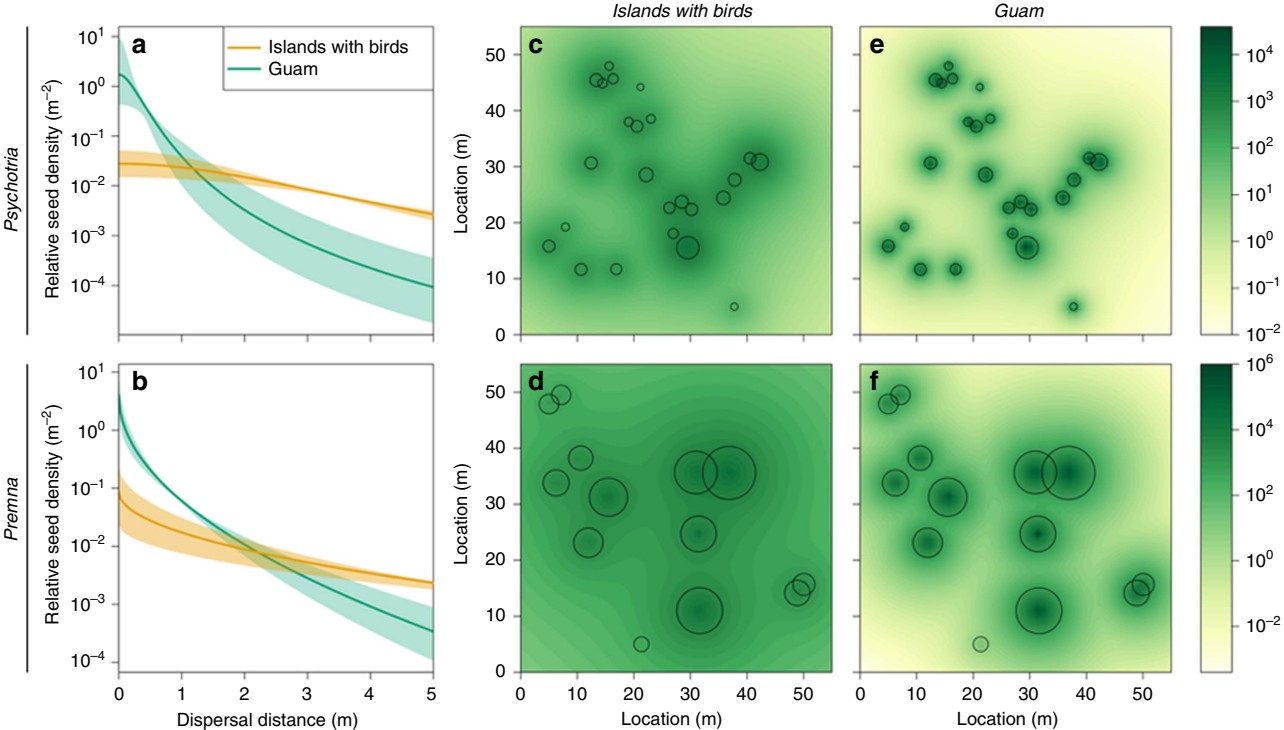

**Figure 3 | Dispersal across the landscape.** Seed dispersal kernels of (**a**) *Psychotria* and (**b**) *Premna* on Guam as compared with three nearby islands where forest birds are present. Curves in **a,b** show the relative density of seeds arriving at a given distance in any direction from the parent tree, normalized by total fecundity (that is, the integral over all distances and directions is 1). Note logarithmic *y*-axis scale. Lines are posterior means and shaded regions are 95% credible intervals for the best-supported kernel (2D*t* for *Psychotria*; power-exponential for *Premna*) estimated using hierarchical Bayesian methods. The fitted kernel models were used to predict seed rain in a hypothetical forest plot, with tree locations and sizes based on one arbitrarily selected study site. Panels **c,d** depict seed rain in forests with frugivores, and panes **e,f** depict seed rain in forests without frugivores. Shading indicates the mean posterior predictive seed density (seeds m$^{-2}$, note logarithmic scale) arriving at each location on the forest floor. Circles indicate crowns of conspecific adult trees (drawn to scale from field measurements).

near conspecific trees (Supplementary Fig. 2, Supplementary Table 8,9). The advantage of dispersal was similar across islands, suggesting that distance-mediated effects are more important than density effects, given that seed density is much higher under parent trees on Guam. In a separate study on germination of *Premna* seeds, Fricke *et al.*[26] found five times greater survival in germination and early seedling survival for individuals far from conspecifics compared with those near conspecifics (Supplementary Table 10, 11). In all, these results demonstrate that few seeds on Guam escape the negative effects associated with proximity to parent plants.

**Integrative recruitment model.** By integrating statistical models of bird impacts on seed ingestion, dispersal, germination, and seedling survival, we calculated the probability that a seed produced by a single, isolated parent tree will survive to become an established seedling with and without birds present. For *Premna*, we predict that the cumulative impact of snake-induced bird loss reduces seedling recruitment by 87–92% (Supplementary Table 12). For *Psychotria* the reductions range from 61 to 72%, but estimates for *Psychotria* are conservative because they only include distance-dependent mortality effects on seedling survival, not on germination (see Methods).

## Discussion

These results paint a bleak picture for diversity in the snake-ridden, bird-free forests of Guam. The large negative impacts of disperser loss on regeneration of these two tree species foreshadows profound impacts on the island's fleshy-fruited tree populations, about 70% of island's tree species. Previous work in this system suggests that bird loss will also slow forest regeneration in disturbed areas due to inadequate seed dispersal[27], with potentially negative impacts on carbon storage[28].

Invasive species with widespread impacts, such as the loss of all frugivores caused by the brown treesnake, can overwhelm resilience provided by redundant mutualistic interactions[29]. However, invasive predators are likely to have indirect effects associated with the decline of their vertebrate prey populations, even without causing extinction. For example, fruit bats in Tonga became ineffective dispersers at intermediate densities because they moved less and spent more time in each tree[30]. Even partial loss of the frugivore community is likely to affect dispersal, given that frugivore species vary in their impacts on germination after handling[11] and their propensity to bring seeds to suitable microsites[31].

On Guam, natural resource managers face significant challenges. Island-wide eradication of the brown treesnake is currently not feasible, making widespread reintroduction of native dispersers still present on nearby islands a challenge. In addition, ecosystem-scale management of forest diversity through manual seed collection and dispersal is unrealistic. However, local snake control through fencing, trapping, and/or toxicants is possible, and could be combined with the reintroduction or expansion of native bird species to make the restoration of ecological function across targeted areas an achievable goal[32,33].

Our study highlights an unrecognized but likely widespread impact of invasive predators. Invasive cats (*Felis silvestris catus*)

have affected over 175 threatened vertebrate species on islands alone[34], and the invasive ship rat (*Rattus rattus*) is associated with the decline or extinction of over 60 vertebrate species[35]; together, cats and rats are connected to 44% of bird, mammal and reptile extinctions since AD 1500 (ref. 36). Although easily overlooked because of the indirect nature of the interaction, mutualistic disruption by invasive vertebrates is likely operating at the global scale with pervasive impacts on plant communities. While we recognize that most non-native species have no negative effect and in some cases can replace function formerly provided by a native species[21], the full impacts of truly invasive species must be explored before we consider them 'Nature's Salvation'[3].

## Methods

**Site description.** This study was conducted in the four largest islands in the Mariana Islands chain: Guam—(541 km$^2$), Saipan (115 km$^2$), Tinian (101 km$^2$) and Rota (85 km$^2$). The four islands are within 120 miles of each other and experience similar temperature and rainfall. The predominant undisturbed forest type (hereafter referred to as limestone forest, which reflects the substrate) contains ∼45 native tree species, with around 70% dispersed primarily by birds. This forest type historically covered >30% of the land area on each of the Mariana islands and houses most of the bird, snail, insect, lizard and bat species in greatest danger of extinction[37,38]. The brown treesnake (*Boiga irregularis*) was introduced to Guam in the mid-1940's and caused the functional extinction or extirpation of all native forest birds[8]. The frugivorous bird fauna on Saipan, Tinian, Rota and Guam includes seven species (Supplementary Table 1), although fruit makes up only a small portion of the diet for one species (Mariana Crow). Only the Micronesian Starling is currently present on Guam, in one small population in the urbanized area of Anderson Air Force Base where snakes are controlled via trapping. The Mariana Fruit Bat is also a native frugivore, but has been functionally or completely extirpated from Guam, Saipan, and Tinian, and is in low abundance on Rota (∼2,000 individuals). The brown treesnake is partially responsible for the functional extirpation of the fruit bat on Guam[39].

**Focal species.** *Psychotria mariana* (family Rubiaceae) is an understory/canopy species with small (6.9 × 7.9 mm) red, fleshy fruits with two seeds per fruit. *Premna serratifolia* (formerly *P. obtusifolia*, family Verbenaceae) is a large canopy tree with small (4.4 × 4.6 mm), purple, fleshy fruits with a single seed in each fruit. The fruits occur in a large umbel. Both species are frequently visited by birds; the golden white-eye, bridled white-eye, Mariana fruit dove, white-throated ground dove, Mariana Crow and Micronesian starling have all been observed eating one or both of these species[40–42].

**Germination experiments.** We conducted a nursery experiment to test whether ingestion by birds affects germination. To collect falling fruit and seeds, we placed seed traps under fruiting trees on Saipan, Tinian and Rota in late April and May 2009. Fruit and seeds were removed from traps bi-weekly, air-dried, and stored in a cool, dry room until ready to use. In early June 2009, we sorted seeds into ingested and whole fruit categories, then divided the whole fruit into two equal groups and randomly selected one of the groups for the manually de-pulped treatment. We removed the flesh from all seeds in this group. We planted whole fruit (n = 117 *Premna*, 155 *Psychotria*), de-pulped fruits (n = 104 *Premna*, 143 *Psychotria*), and ingested seeds (n = 150 *Premna*, 173 *Psychotria*) in individually labelled cells. All seeds and fruits were planted under 60% shadecloth at an outdoor nursery on Guam. Plants were watered and checked for germination daily from June to October 2009. Trays were monitored until mid-December, but no new germinants were seen after October, so we assume all viable seeds had germinated.

We used a binomial GLM to test whether germination probability was affected by treatment (whole fruit, seed with flesh manually removed, ingested seed). To determine whether treatment was an important predictor of germination, we used AIC$_c$ values to compare models with and without the fixed effect of treatment. Given that the best-fitting model included treatment, we determined whether the likelihood of germination differed between treatments by using profile likelihoods to estimate 95% confidence intervals around the coefficients first using contrasts with 'whole fruit' as the reference level, and then using contrasts with 'flesh manually removed' as the reference level. If the confidence intervals did not include zero, we concluded that the likelihood of germination of seeds within that treatment differed from the reference level. The two species (*Psychotria* and *Premna*) were analysed separately.

**Seed dispersal kernels.** To measure the seed shadow of trees on Guam relative to trees on nearby islands with birds (Saipan, Tinian and Rota), we set up seed traps on all four islands. We selected fruiting focal trees by going to a randomly selected GPS point (assigned using random point generator in ArcGIS) in high-quality forest, searching a 30 m radius for all conspecific trees, and selecting the largest tree for the focal tree. We set up a

wedge-shaped array of 17 seed traps radiating from the focal tree in order to ensure a variety of distances away from fruiting trees were sampled, with a particular emphasis on near distances which might not be captured in a random trap array. Although the focal tree was a dominant tree of that species in the forest, other conspecific trees also contributed to the seed traps, therefore each array sampled seed rain from multiple trees. We established four arrays on Guam and three on each of the other islands, giving a total of 13 seed trap arrays per species (221 traps). Conspecific arrays were located at least 500 m and often several km apart. All conspecific trees within 20 m of any seed trap were measured (diameter at breast height) and assigned UTM coordinates using a high-accuracy Trimble GPS. Our modelling approach considered all trees of the focal species as potential contributors to seed traps. See analysis section below for details on modelling dispersal kernels.

Seed traps were made by forming flexible PVC tubing into a hoop, then securing screen door mesh from the hoop in a bowl shape. Traps were hung at 1.3 m by rope attached to neighbouring trees. Traps were maintained for 10 months (April 2008–January 2009) for *Premna* and for 4 months (April–July 2009) for *Psychotria*, which includes one entire annual fruiting season for each species. Seed trap contents were collected every 4–6 weeks, dried and sorted. All seeds from the focal species of interest for each trap were counted. Seeds in each trap were summed across all months of collection.

Recent developments in quantitative ecology provide a powerful and flexible approach to the problem of inferring the dispersal of tree propagules (for example, seeds or fruits; hereafter we simply refer to 'seeds') from measurements under a closed canopy, with multiple potential propagule sources[13,43–45]. The heart of the approach is a model of the 'seed shadow' from a single parent tree as the product of (1) total seed production Q over a defined time interval and (2) the dispersal kernel *f*(*r*,ω), a two-dimensional probability density function (pdf) which gives the normalized density of seeds arriving at a radial distance *r* from the parent in a direction specified by angle ω:

$$s(r,\omega) = Qf(r,\omega). \qquad (1)$$

Assuming directional isotropy, the seed shadow *s*(*r*), in units of seeds m$^{-2}$, is independent of ω.

Total seed production or 'source strength' is often assumed to be proportional to tree basal area *b* (but see ref. 46), and we follow this convention here. Our model includes stochastic variation in the size-fecundity relationship at the site level, recognizing that observations are spatially grouped by site (and hence avoiding pseudoreplication) and that sites may differ in environmental factors that affect fecundity[47]. For tree *i*,

$$Q_i = \beta_s b_i \qquad (2)$$

$$\log(\beta_s) \sim N(\mu_\beta, \sigma_\beta), \qquad (3)$$

where the subscript *s* denotes the site at which tree *i* is located and the site-level random effects are lognormally distributed with common log-mean $\mu_\beta$ and log s.d. $\sigma_\beta$.

Various parametric dispersal kernel models have been proposed, differing in characteristics such as the degree of kurtosis and curvature near the source[13,45]. We considered two of the kernel forms most frequently used in studies of seed dispersal, the power-exponential[43,44] and 2D*t* (ref. 45):

$$f(r) = \frac{p}{2\pi a^2 \Gamma(\frac{2}{p})} \exp\left[-\left(\frac{r}{a}\right)^p\right] \qquad \text{power} - \text{exponential} \qquad (4)$$

$$f(r) = \frac{p}{\pi a \left[1 + \left(\frac{r}{a}\right)^2\right]^{p+1}} \qquad \text{2D}t \qquad (5)$$

In both cases, *a* is a scale parameter (units of m), with larger values of *a* 'stretching' the kernel to allow longer dispersal, and *p* is a nondimensional shape parameter defined such that smaller values of *p* correspond to fatter-tailed kernels (that is, higher probability of long-distance dispersal). Integrating the kernel around a circle of radius *r* (that is, multiplying equations 4 and 5 by 2π*r*) gives the marginal pdf of radial dispersal distance *r*, from which statistics such as the mean, mode and quantiles of dispersal distance can be derived[45]. Like other authors[24,43,44], we found that the shape parameter was not well-identified by the data (see description of estimation methods below), so we fixed it at *p* = 0.5 for the power-exponential (that is, the 'exponential square-root') and *p* = 1 for the 2D*t*. Although seed dispersal, like fecundity, is a complex process that varies at multiple scales[48], we do not attempt to pursue hierarchical models for the kernel and instead focus on the effect of birds on dispersal ability. The key question for our analysis is whether the kernel scale differs between Guam (without birds) and Rota, Saipan and Tinian (with birds). We formalize this by comparing the support, given the data, for three hypotheses about the vector of island-specific scale parameters **a** = [$a_G$, $a_R$, $a_S$, $a_T$]: (1) no differences among islands ($a_l$ = *a* for all *l*), (2) Guam differs from the other islands ($a_G ≠ a_R = a_S = a_T$) or (3) each island is distinct ($a_G ≠ a_R ≠ a_S ≠ a_T$).

The parameters in equations 1–5 can be estimated using seed traps placed in a mapped stand by comparing the observed number of seeds collected in trap *j* at site *s* on island *l* to the predicted number, where the latter is found by summing the seed shadows across all $N_s$ potential parent trees at site *s*, given their distances to

the trap and the trap area $A$:

$$\hat{s}_j\left(\mathbf{b}, \mathbf{r}_j | \beta_s, a_l\right) = A \sum_{i=1}^{N_s} s(b_i, r_{ij} | \beta_s, a_l). \tag{6}$$

Equation 6 forms the basis for a likelihood function of the data given the parameters. Because seed rain is often clumped, especially for animal-dispersed taxa[44,49], we used a negative binomial (NB) likelihood to capture overdispersion of observed seed counts around their fitted means. Our parameterization of the NB uses two parameters, $k_1$ and $k_2$, to describe the mean-variance relationship as a second-degree polynomial[50], which substantially improved the fit to the data over the standard parameterization. The total likelihood of the data is the product of the pointwise likelihoods over all traps at all sites.

We used a hierarchical Bayesian framework to fit the models[25]. The posterior probability distribution of the parameters given the data (that is, the vector of seed counts $\mathbf{s}$) is

$$p(\boldsymbol{\beta}, \mathbf{a}, \mu_\beta, \sigma_\beta, k_1, k_2 | \mathbf{s}) \propto p(\boldsymbol{\beta}, \mathbf{a}, \mu_\beta, \sigma_\beta, k_1, k_2) \times \prod_{s=1}^{13} N(\log \beta_s | \mu_\beta, \sigma_\beta) \times \prod_{j=1}^{221} \mathrm{NB}(s_j | \boldsymbol{\beta}, \mathbf{a}), \tag{7}$$

where the first factor is the prior probability of the parameters. We used independent, noninformative priors. For $\sigma_\beta$, $k_1$ and $k_2$, the prior was uniform over the non-negligible range of posterior density; for $\mu_\beta$ and all $\log(a_j)$ the prior was a diffuse normal with mean zero and variance $10^6$. We simulated random draws from the posterior distribution using Markov chain Monte Carlo[51], implemented in JAGS run from R using the R2jags package[52]. After an initial burn-in period, 1,000 samples were retained from each of three parallel chains, using a suitable thinning interval to reduce within-chain autocorrelation. Convergence was assessed using traceplots, histograms, and the Gelman–Rubin diagnostic. For each species, we fit six candidate models (two kernel forms × three models of inter-island scale differences). We used the deviance information criterion[53] to compare the strength of evidence among models within a species.

**Proportion of seeds ingested.** Seeds collected in seed traps were categorized as 'ingested' or 'uningested', depending on whether the seed was bare or covered by fleshy fruit skin. While frugivores are the primary cause of flesh removal, weathering and ants may be partially responsible, but would likely affect fruit on all islands similarly. To test whether the proportion of seeds ingested differed between Guam (no birds) and islands with birds, we used generalized linear mixed effects models (GLMMs) with a binomial error distribution. Collection site was included as a random effect on the intercept. To determine whether the fixed effect of island was an important predictor of the proportion of seeds ingested, we used $AIC_c$ values to compare models with and without the fixed effect of island. Given that the best-fitting model includes island, we determined whether the proportion of seeds differed between each island with birds (Saipan, Tinian and Rota) and Guam by using profile likelihoods to estimate 95% confidence intervals around the coefficients, based on contrasts with Guam as the reference level. If the confidence intervals did not include zero, we concluded that the proportion of seeds handled on that island differed from the proportion handled on Guam. The two species (Psychotria and Premna) were analysed separately.

To determine whether the proportion of seeds that are ingested differs with respect to distance from the nearest conspecific, we again used a GLMM with a binomial error distribution. We only used data from Saipan, Tinian and Rota, since very few seeds were found in seed traps away from the parent tree on Guam. The proportion of seeds ingested was used as the response variable and island, distance to nearest conspecific and an island by distance interaction were potential predictors. Collection site was included as a random effect on the intercept. We fit the full model (island by distance interaction) and then all submodels (island + distance, island alone, distance alone, null model), and used model comparison via $AIC_c$ values to identify the best-fitting model. The two species (Psychotria and Premna) were analysed independently.

**Seedling stage distance-dependent mortality experiment.** We collected ripe fruits and seeds from Premna and Psychotria trees on each island. Seedlings were grown in nurseries on Saipan (for Saipan and Tinian plants), Rota and Guam in September 2010. When seedlings grew their first true leaves, we transplanted the healthiest seedlings into single cell trays, then outplanted them into the field 1–3 months later.

Outplanting occurred at five sites on Guam and three sites each on Saipan, Tinian and Rota; all sites were at least 500 m and often several km apart, and nearly all sites were the same as or adjacent to the sites used to collect seed rain (above). Sites were selected, based on the following criteria: must be comprised primarily of native tree species, have a karst substrate, and be dominated by Ficus spp. and Pisonia grandis, with an understory of Guamia mariannae, Aglaia mariannensis, Cynometra ramiflora and/or Eugenia reinwardtiana. Within a 60 × 60 m area at each site, we mapped the location of all adult (>4 cm diameter at breast height) trees, and used these maps to select locations that were either near (under or within 1 m of the canopy) or far (typically >7 m) from the parent tree. If there were few adult trees of the same species, we placed two to four 'near' plots under the same

tree. We set up a chicken wire fence (height 0.9–1.2 m) around each plot to prevent browsing by pigs or deer. These ungulates are present only on Guam and Rota, but the fences were added to plots on Saipan and Tinian as well to control for any fence effect.

We planted 10 seedlings in each near and far plot designated for that species at each site on all four islands. Eighty seedlings of each species were planted per site for a total of 1,120 Premna and 700 Psychotria seedlings across all islands. Premna was planted at every site on all islands (total 14 sites). Psychotria was planted at three sites on Saipan, three sites on Rota and three sites on Guam (total nine sites). We did not outplant Psychotria on Guam due to a lack of adults or at any sites on Tinian due to a lack of seedlings. Premna was outplanted in December 2010, and Psychotria between January 2011 and March 2011. The vast majority of plots, near or far, contained zero conspecific seedlings and few plots contained more than one or two conspecific seedlings, therefore we did not manipulate existing conspecific seedling density or include it in the analysis. Premna seedlings were surveyed for survival in July 2011, and Psychotria seedlings were surveyed in July 2012.

Because light limitation has a strong effect on seedling survival, we used a spherical densiometer to record canopy openness at each fence. Four measurements were taken per fence (one in each cardinal direction), then averaged to get a single estimate of canopy openness. single individual conducted all densiometer readings, a All sites on an island were surveyed in the same day, and surveys were not conducted during strong wind or rain showers.

We used a binomial GLMM to test the impacts of distance to the nearest conspecific adult ('near' versus 'far') on the proportion of seedlings alive in each plot at the end of the experiment. The full model included bird presence, nearest-neighbour distance and their interaction as fixed effects to test whether any change in survival with distance was steeper on Guam. We also included canopy openness and the openness × distance interaction to test whether the effect of distance was modified by light environment. Canopy openness values were mean-centered before analysis. All models included site as a random effect on the intercept. We used model comparison based on $AIC_c$ to identify the best-fitting model. We used profile likelihood to estimate 95% confidence intervals around the coefficients, based on contrasts with Guam (no birds) and 'near' as the reference levels for the island and distance factors respectively. The two species (Psychotria and Premna) were analysed independently.

**Seed-to-seedling stage distance-dependent mortality experiment.** We quantified distance-dependent mortality at germination and early seedling stages for Premna using seed additions near and far from conspecific adults on Saipan. These experiments are described in detail in Fricke et al.[26]. Briefly, small (0.05 m²) plots were placed at six near and six far locations at each of three mapped forest plots, each at least 500 m apart, giving a total of 18 plots per distance category. Seeds were collected from >10 individuals, fruit pulp was manually removed, and seeds were sown at a density of 50 seeds per plot (total of 900 seeds sown per distance category). Seed addition plots were partially surrounded with wire mesh to maintain a consistent environment with other experimental manipulations performed concurrently for a separate project, but large holes in the mesh allowed entrance to all natural enemies. We recorded the total number of germinants using weekly checks, and assessed seedling survival five weeks after germination began. We recorded canopy openness above each plot using a spherical densiometer. To analyse the effect of distance on germination and early seedling survival we used GLMs with a binomial error distribution. The proportion of seeds germinated or surviving by the end of the study period was used as the response variable, and each stage was tested separately. Distance, canopy openness, the interaction between distance and canopy openness, and site were included as fixed effects. We fit the full model and all submodels, and used model comparison via $AIC_c$ values to identify the best-fitting model.

**Integrative metric showing impact of bird loss.** To quantify the cumulative impact of bird loss across the early life history stages of the two study species, we combined the empirically parameterized models described in the preceding sections. The probability that a seed survives through the seedling stage depends on the probability $p(r)$ of dispersing to a distance $r$ from the parent (given by the marginal distance pdf derived from the two-dimensional dispersal kernel shown in equations 4 and 5), the probability $p(I|r)$ that the seed was ingested by birds given its dispersal distance, the probability $p(G|I, r)$ of germination (which depends on whether the seed was ingested and on dispersal distance), and finally the early and late seedling survival probabilities, $p(S_1|r)$ and $p(S_2|r)$ respectively, given distance-dependent effects. In the interest of constructing a simple metric, we ignore the effects of neighbouring conspecific adult canopies by asking how the survival rate of a seed produced by a single, isolated parent tree varies depending on bird presence or absence. Integrating across dispersal distance, overall seed-to-seedling survival is

$$p(\text{seed} \rightarrow \text{seedling}) = \int_0^\infty p(r)[p(I|r)p(G|I, r) + (1 - p(I|r))p(G| \sim I, r)] \tag{8}$$
$$p(S_1|G, r)p(S_2|G, r)\mathrm{d}r.$$

As described above, data and models for early seedling survival and distance-dependent effects on germination are available for *Premna*[26] but not for *Psychotria*. Because we cannot estimate absolute seed-to-seedling survival for *Psychotria*, we focus instead on the ratio of survival in the absence versus in the presence of birds for both species. We note that the comparison is likely conservative (that is, the ratio is likely biased high) in the case of *Psychotria* if this species does in fact experience positive distance-dependence in germination or early seedling survival.

By making explicit the dependence of each transition probability on a parameter vector $\theta$ (which includes kernel shape and scale, as well as the regression coefficients that define the generalized linear models for ingestion, germination and seedling survival), the expected seed-to-seedling survival can be calculated as

$$E[p(\text{seed} \rightarrow \text{seedling})] = \int p(\text{seed} \rightarrow \text{seedling} |\theta) p(\theta| \text{Data}) d\theta. \quad (9)$$

We propagated uncertainty as shown in Equation 9 by randomly simulating kernel scale from its posterior distribution and generating regression coefficients from their multivariate normal sampling distributions using the covariance matrix of the fixed-effect MLEs for each generalized linear (mixed) model. For all transitions, we used the best-supported model (based on DIC or $AIC_c$) for a given species. We did not include any random effects in the calculations since we were interested in average, island-level predictions. For seedling survival models that include canopy openness as a covariate, we generated predictions using the average openness across all fenced plots on all islands. For the 'birds present' scenario, we applied the kernel scale estimate for islands with birds and the island-specific regression parameters for ingestion probability (except in the case of Guam, where no seeds are ingested because birds are absent, so simulated values were taken to be the average of the values predicted for the other three islands). For the 'birds absent' scenario, we used the kernel scale estimate for Guam and set ingestion probability to zero. To estimate $p(G|I,r)$ for *Premna*, we combined terms from the separate binomial GLMs fitted to the greenhouse germination data (which estimated the effect of ingestion) and the field germination data (which estimated the effect of dispersal distance). Specifically, we used the greenhouse GLM as a baseline and added a distance effect with the coefficient simulated from the field GLM to predict the log-odds of germination for ingested or uningested seeds at a given dispersal distance. For *Psychotria*, no data on distance-dependent germination or early seedling survival were available, so we simply used the greenhouse GLM to estimate $p(G|I)$ and treated $p(S_1)$ as constant. For each random set of parameters, we evaluated equation 8 by numerical integration under both scenarios, using identical values for parameters that are shared between the scenarios. Taking the ratio of these paired realizations, we calculated means and quantiles of the resulting vector of predicted relative seed-to-seedling survival values as the metric of bird loss impact.

All statistical analyses were performed using R (ref. 54). The lme4 package was used for GLMMs[55].

**Code availability.** Analysis scripts will be available on GitHub at https://github.com/EBL-Marianas/MutualismDisruption_NatureComm upon publication.

**Data availability.** The seed to seedling transition distance-dependent mortality experiment data is at: doi:10.5061/dryad.8qg42. Other data are available from the corresponding author on request.

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

## Acknowledgements

Funding was provided by the Budweiser Conservation Scholarship through the National Fish and Wildlife Foundation, the National Science Foundation Graduate Research and IGERT Fellowships to HSR, and the National Science Foundation DEB-0816465 and US Department of Agriculture National Research Initiative 2008-03106. Thanks to Kaitlin Mattos, Eliza Hooshiar, Isaac Chellman and the EBL field crew for data collection.

## Author contributions

The study was conceived and designed by H.S.R., J.H.R.L., R.H.M. and J.J.T.; H.S.R., E.C.F. and field assistants collected the field data; H.S.R. and E.R.B. analysed the data; the manuscript was written by H.S.R., with input from J.H.R.L., J.J.T. and E.R.B., and comments provided by R.H.M. and E.C.F.

## Additional information

**Competing financial interests:** The authors declare no competing financial interests.

