## [Peer Review File · Nature Communications]

Reviewers' Comments:

Reviewer #1 (Remarks to the Author):

This is a wonderful paper. It identifies a key indirect effect - loss of dispersal agents for trees - caused by a widely known direct effect - elimination of vertebrates by invasive competitors or in this case predators. I will use this paper in the fall in a class if it is out in time, and certainly cite it repeatedly. My comments below are to help make it more digestible to a general audience, as well as anchor it in some important literature that might be consulted for citation.

What you are testing as the mechanism for recruitment failure in the trees is one of the Janzen-Connell effects. The distance effect was stressed more by Janzen, so that paper should be cited. There are now something like 600-800 papers published a year on some aspect of seed dispersal. Flaky it is, but that would not have happened without Janzen (1970 Am. Nat.).

A general comment is that the abstract and narrative before methods flow well and are almost certainly correct. Two points might help.

The analytical framework is sophisticated but complex. There are people who will appreciate that. There are also people who will be troubled by all the potential hidden interactions of so many uncontrolled variables. There are statistically competent ecologists who will be impressed and comfortable with the approach, professional statisticians who will get hives thinking about any field data, and everyone else. I do not recommend re-doing analyses, but you may want to deflect some of those worries with some simple observations. Are there seedlings and saplings under fruiting trees on Guam? Given the number of seeds without flesh within 5 m of the trees on the other islands, are there seedlings, juveniles and saplings there? I searched for "sapling" in the paper and did not get a hit. Even a general statement that recruitment of seedlings, juveniles and saplings under reproductive trees is rare would help make the argument that dispersal is important. Do you have a picture? Is there even a back of the envelope way of estimating how many seeds each tree produces, and how many survive to a later stage nearby? Then your analysis is finding the cause through lack of dispersal what I expect is a devastating effect on Guam. That kind of thinking is what made Janzen (1970) plausible and stick, long before there was any real evidence. For those who have worked with seedlings in the field, projections of vitals from constrained greenhouse studies are big jumps.

The second general issue is one of context. A very few sentences here and there could help. For

instance, negative density dependence is almost certainly general, but has many mechanisms. Seeds close to the mother or other conspecifics might share enemies, seedlings might, juveniles might, and saplings or poles might. Those agents may be soil organisms, insects, disease above ground, mammalian herbivores, or even birds foraging under trees that kill seeds rather than disperse them. One paper you cited (Harms et al. 2000 Nature) found some density dependent mortality in transition from seeds to seedlings in all 53 species that had real numbers. Colleagues confirmed it with hundreds more traps later. You have identified distance as critical, but in some minds have not excluded density effects, partly because of the mess of possible effects of whole fruit and seeds under parents, partly because in some species seedling or sapling effects come later. Following seeds and seedlings in 20x25 cm plots for five weeks is suggestive, but not the whole story. You mentioned that seedlings may not do well near parents because of shared enemies. Mention the multitude of possible reasons, and cite some experiments that show that your soil inoculations make sense (Mangan et al. 2010 Nature).

That being said, as long as your species are not proliferating around fruiting adults on Guam your conclusions are almost certainly correct.

It might have been there, but I didn't see it. One of Traveset's important contributions was that different fruit-eating birds have different effects on germination of the same tree species (your citation #11 and elsewhere). If there are no birds on Guam, your basic argument is correct. But it would help to acknowledge that birds on the other islands may have quite different effects (consistent with citation #11), and influence the dispersal kernels. In that light, McConkey and Drake (2006 Ecology) make the point that dispersal function of fruit-eating bats on Pacific islands disappears long before the critters are extinct. Mentioning it would strengthen your argument.

Figure 1 needs to identify the birds in the caption. For those in the Americas and Europe who are used to thinking of pigeons and doves as lethal to seeds because they grind them up in gizzards, it would not hurt to mention in the text that fruit doves/pigeons are effective dispersal agents throughout Asia and the Pacific. Is the ground dove a consistent dispersal agent?

Finally, some fine tuning on the general issue might help. I doubt if the wildest-eyed true believer in inevitability of novel ecosystems would categorize brown tree snakes as advantageous to anything but themselves, and Malthus might even take over for them. It is legitimate to present that case that some invaders are catastrophic. It would soften the potential rage to mention that many exotics are not. In highly disturbed tropical ecosystems - somewhere between 50-70% of the land area of the tropics - conserving pristine interactions is a moot point. Invasives are there, some provide services, perhaps sufficient to allow communities to stabilize around a different norm than we know, but they don't destroy whole functional groups of organisms. A thoughtful essay by a number of people with insight is McConkey et al. (2012 Biological Conservation).

This might not be useful for the present paper, but can you see any way that brown tree snakes can be controlled? As you suggest, the novel ecosystem is on Guam, and it is depauperate and becoming more so. A little tweaking might drive that point home.

Anyway, this is really good work. It shines a bright light on a really clear example. Undergrads in my courses will know about it as soon as it is citable.

Reviewer #2 (Remarks to the Author):

I enjoyed reading this paper. As the authors correctly state, INDIRECT impacts of invasive species are seriously underrated when assessments are made of the implications of biological invasions. There is an urgent need to expand the assessment of the mechanisms whereby invaders affect invaded ecosystems (Kumschick et al. 2015 *BioScience* 65: 55-63). As Traveset & Richardson (2011) have argued, mutualisms are a key casualty of invasions (but an understudied one).

This study utilizes one of the best possible study systems available to explore the implications of the loss of services provided by frugivores on the regeneration of forest tree species. I read the methods and results carefully and find these to be well written, clear and accurate. The interpretation of results appears sound to me. I found the results to be convincing and conclude that the paper provides a very important contribution to the literature on impacts of invasions and a call to give more careful attention to indirect effects when quantifying overall impacts.

I congratulate the authors on a very good piece of work.

Reference cited

Traveset, A. & Richardson, D.M. (2011). Mutualisms - key drivers of invasions... key casualties of invasions. In: Richardson, D.M. (ed.) *Fifty years of invasion ecology. The legacy of Charles Elton*, pp. 143-160. Wiley-Blackwell, Oxford

Reviewer #3 (Remarks to the Author):

This paper compares early life history stages of two tree species dispersed by birds in Guam, an island where most frugivore birds have gone extinct by predation by an introduced snake, and three adjacent islands with these frugivores present. They found more seed rain of viable seeds

and longer seed dispersal (far from canopy trees) in non-invaded islands. Seed germination increased after bird gut passage and seedling survival decreased close to canopy trees. These results suggest lower seed survival in Guam compared to the other islands as an indirect consequence of bird dispersal loss by snake predation.

At first I was worried with only one "island" treatment replicate, but invasion by the Guam snake is a "natural experiment" that luckily has so far not occurred in other regions, therefore it is impossible to have more replicates; yet I'd like to make sure that the experiments were repeated widespread across the island, at a geographical range were besides the loss of seed dispersers, there is environmental variation. One more issue of concern is the lack of information with regard population tree density and specially seedling density at those sites. These values are important to show and compare between invaded and non-invaded islands. Has this decrease in seed dispersal translated to species demographic effects? The authors should also frame their study in relation to defaunation. There are many studies on defaunation consequences on seed dynamics.

Details:

Fig 2 legend. Refer to e) and f).

Fig 3. Not clear why the position of trees shown as case studies is the same in Guam compared to the other islands. This is not at all possible.

Response to Reviewers' Comments on NCOMMS-16-13124-T

We have responded to all reviewer comments below, and adjusted the text in accordance with nearly all of their suggestions. Our responses are in italics following an asterisk below.

Reviewer #1 (Remarks to the Author):

This is a wonderful paper. It identifies a key indirect effect - loss of dispersal agents for trees - caused by a widely known direct effect - elimination of vertebrates by invasive competitors or in this case predators. I will use this paper in the fall in a class if it is out in time, and certainly cite it repeatedly. My comments below are to help make it more digestible to a general audience, as well as anchor it in some important literature that might be consulted for citation.

**Thank you!*

What you are testing as the mechanism for recruitment failure in the trees is one of the Janzen-Connell effects. The distance effect was stressed more by Janzen, so that paper should be cited. There are now something like 600-800 papers published a year on some aspect of seed dispersal. Flaky it is, but that would not have happened without Janzen (1970 Am. Nat.).

**This citation has been added to P3 L17.*

A general comment is that the abstract and narrative before methods flow well and are almost certainly correct. Two points might help.

The analytical framework is sophisticated but complex. There are people who will appreciate that. There are also people who will be troubled by all the potential hidden interactions of so many uncontrolled variables. There are statistically competent ecologists who will be impressed and comfortable with the approach, professional statisticians who will get hives thinking about any field data, and everyone else. I do not recommend re-doing analyses, but you may want to deflect some of those worries with some simple observations. Are there seedlings and saplings under fruiting trees on Guam? Given the number of seeds without flesh within 5 m of the trees on the other islands, are there seedlings, juveniles and saplings there? I searched for "sapling" in the paper and did not get a hit. Even a general statement that recruitment of seedlings, juveniles and saplings under reproductive trees is rare would help make the argument that dispersal is important. Do you have a picture? Is there even a back of the envelope way of estimating how many seeds each tree produces, and how many survive to a later stage nearby? Then your analysis is finding the cause through lack of dispersal what I expect is a devastating effect on Guam. That kind of thinking is what made Janzen (1970) plausible and stick, long before there was any real evidence. For those who have worked with seedlings in the field, projections of vitals from constrained greenhouse studies are big jumps.

**Indeed, there are few seedlings or saplings of these two species on Guam, and if any seedlings are found, they tend to be under fruiting trees. There are more seedlings on islands with dispersers, both near and far from parent trees. However, the reason we did not simply compare*

seedling abundance is that there are invasive deer on Guam and Rota that likely reduce seedling abundance under parent trees. We have added the following sentence to P4 L6-10:

“Seedlings and saplings of both species are rare on Guam relative to nearby islands, consistent with the prediction that disperser loss reduces plant recruitment. However, the abundance of introduced ungulate herbivores is also highest on Guam, which makes it difficult to attribute recruitment failure solely to mutualism disruption. Therefore, we designed our study to isolate the role of dispersers.”

The second general issue is one of context. A very few sentences here and there could help. For instance, negative density dependence is almost certainly general, but has many mechanisms. Seeds close to the mother or other conspecifics might share enemies, seedlings might, juveniles might, and saplings or poles might. Those agents may be soil organisms, insects, disease above ground, mammalian herbivores, or even birds foraging under trees that kill seeds rather than disperse them. One paper you cited (Harms et al. 2000 Nature) found some density dependent mortality in transition from seeds to seedlings in all 53 species that had real numbers. Colleagues confirmed it with hundreds more traps later. You have identified distance as critical, but in some minds have not excluded density effects, partly because of the mess of possible effects of whole fruit and seeds under parents, partly because in some species seedling or sapling effects come later. Following seeds and seedlings in 20x25 cm plots for five weeks is suggestive, but not the whole story. You mentioned that seedlings may not do well near parents because of shared enemies. Mention the multitude of possible reasons, and cite some experiments that show that your soil inoculations make sense (Mangan et al. 2010 Nature).

**We appreciate this suggestion, and have added a few details regarding the possible mechanisms causing distance or density-dependent mortality (P6 L11-13). In terms of our distance dependence experiments, we followed seedlings for 5-weeks post-germination in the field in one study, and we followed approximately 3-month old seedlings for another 5-8 months in a second field study. Neither study was in a greenhouse or used soil inoculations. Since they were in the field, all possible natural enemies could have influenced their survival relative to distance. Rather than trying to isolate the individual effects of density and distance, we recognize that disperser loss will both reduce the distance seeds travel and increase the density of seeds underneath parent trees and either factor could lead to recruitment failure.*

That being said, as long as your species are not proliferating around fruiting adults on Guam your conclusions are almost certainly correct.

**This is indeed the case. We hope the added sentence on P4 L6-10 (copied above) helps readers understand this.*

It might have been there, but I didn't see it. One of Traveset's important contributions was that different fruit-eating birds have different effects on germination of the same tree species (your citation #11 and elsewhere). If there are no birds on Guam, your basic argument is correct. But it would help to acknowledge that birds on the other islands may have quite different effects (consistent with citation #11), and influence the dispersal kernels. In that light, McConkey and Drake (2006 Ecology) make the point that dispersal function of fruit-eating bats on Pacific

islands disappears long before the critters are extinct. Mentioning it would strengthen your argument.

**Great point! We have added both suggestions – that species vary in their effects (P7L23-P8 L2), and that function is lost before species are rare (P7 L20-23)- to the conclusion.*

Figure 1 needs to identify the birds in the caption. For those in the Americas and Europe who are used to thinking of pigeons and doves as lethal to seeds because they grind them up in gizzards, it would not hurt to mention in the text that fruit doves/pigeons are effective dispersal agents throughout Asia and the Pacific. Is the ground dove a consistent dispersal agent?

**We have added the species names in the caption to Figure 1. We recently conducted feeding trials on Saipan with most of these bird species and will continue to monitor germination of gut passed seeds in the coming months. The ground dove appears to be both a predator and a disperser, depending on the species and size of the seed. We have decided to keep the ground dove in the manuscript as a potential disperser, as we know this species consumes the fruits of both species and disperses some species at least some of the time. That said, this decision does not qualitatively or quantitatively change the narrative because in the recruitment model we use data from observed dispersal kernels developed using seed rain data, not data on seed consumption by specific frugivores.*

Finely, some fine tuning on the general issue might help. I doubt if the wildest-eyed true believer in inevitability of novel ecosystems would categorize brown tree snakes as advantageous to anything but themselves, and Malthus might even take over for them. It is legitimate to present that case that some invaders are catastrophic. It would soften the potential rage to mention that many exotics are not. In highly disturbed tropical ecosystems - somewhere between 50-70% of the land area of the tropics - conserving pristine interactions is a moot point. Invasives are there, some provide services, perhaps sufficient to allow communities to stabilize around a different norm than we know, but they don't destroy whole functional groups of organisms. A thoughtful essay by a number of people with insight is McConkey et al. (2012 Biological Conservation).

**We have edited the final sentence (P8 L16-19) to recognize the diverse roles non-native species can play in an ecosystem. It now reads:*

While we recognize that most non-native species have no negative effect and in some cases can replace function formerly provided by a native species, the full impacts of truly invasive species must be explored before we consider them "Nature's Salvation".

This might not be useful for the present paper, but can you see any way that brown tree snakes can be controlled? As you suggest, the novel ecosystem is on Guam, and it is depauperate and becoming more so. A little tweaking might drive that point home.

**We created the second to last paragraph (P8 L3-9) to depict a more optimistic future for Guam, with a nod to some of the recent developments in toxicants, traps, and fences which enable local snake control. It now reads:*

“On Guam, natural resource managers face significant challenges. Island-wide eradication of the brown treesnake is currently not feasible, making widespread reintroduction of native dispersers still present on nearby islands a challenge. In addition, ecosystem-scale management of forest diversity through manual seed collection and dispersal is unrealistic. However, local snake control through fencing, trapping, and/or toxicants is possible, and can be combined with the reintroduction or expansion of native bird species making the restoration of ecological function across a wider landscape an achievable goal.”

Anyway, this is really good work. It shines a bright light on a really clear example. Undergrads in my courses will know about it as soon as it is citable.

**thank you!*

Reviewer #2 (Remarks to the Author):

I enjoyed reading this paper. As the authors correctly state, INDIRECT impacts of invasive species are seriously underrated when assessments are made of the implications of biological invasions. There is an urgent need to expand the assessment of the mechanisms whereby invaders affect invaded ecosystems (Kumschick et al.2015 BioScience 65: 55-63). As Traveset & Richardson (2011) have argued, mutualisms are a key casualty of invasions (but an understudied one).

This study utilizes one of the best possible study systems available to explore the implications of the loss of services provided by frugivores on the regeneration of forest tree species. I read the methods and results carefully and find these to be well written, clear and accurate. The interpretation of results appears sound to me. I found the results to be convincing and conclude that the paper provides a very important contribution to the literature on impacts of invasions and a call to give more careful attention to indirect effects when quantifying overall impacts.

I congratulate the authors on a very good piece of work.

**Thank you!*

Reference cited

Traveset, A. & Richardson, D.M. (2011). Mutualisms - key drivers of invasions... key casualties of invasions. In: Richardson, D.M. (ed.) Fifty years of invasion ecology. The legacy of Charles Elton, pp. 143-160. Wiley-Blackwell, Oxford

Reviewer #3 (Remarks to the Author):

This paper compares early life history stages of two tree species dispersed by birds in Guam, an

island where most frugivore birds have gone extinct by predation by an introduced snake, and three adjacent islands with these frugivores present. They found more seed rain of viable seeds and longer seed dispersal (far from canopy trees) in non-invaded islands. Seed germination increased after bird gut passage and seedling survival decreased close to canopy trees. These results suggest lower seed survival in Guam compared to the other islands as an indirect consequence of bird dispersal loss by snake predation.

At first I was worried with only one "island" treatment replicate, but invasion by the Guam snake is a "natural experiment" that luckily has so far not occurred in other regions, therefore it is impossible to have more replicates; yet I'd like to make sure that the experiments were repeated widespread across the island, at a geographical range were besides the loss of seed dispersers, there is environmental variation.

** We focused on the northern half of the island where limestone forest is the predominant land cover type, but our sites are spread across the northern half. Our field sites were located at least 500 m and often several km apart on Guam. We mention this in the main text in the seed dispersal kernel section (P5 L19), the methods sections under "seed dispersal kernels" (P11 L16-17), and the methods sections for distance-dependent mortality (P16 L13-15).*

One more issue of concern is the lack of information with regard population tree density and specially seedling density at those sites. These values are important to show and compare between invaded and non-invaded islands. Has this decrease in seed dispersal translated to species demographic effects?

**The relatively recent defaunation (1980's) and long lived tree species means that the effects on adult tree density may still be limited at this point – ongoing work with forest mapping will be able to tackle this question. For this study we focus on the consequences on the early life stages most affected by dispersal, where we expect to see large changes. Indeed, we see reduced regeneration on Guam, however, in line with our response to Reviewer 1 on this point, the effects of introduced ungulates may exacerbate the effects of disperser loss on Guam, so we hesitate to do any island to island comparisons. However, by using experiments and observations to disentangle these effects, we can better ascribe population reductions to bird loss. We added a sentence as suggested by reviewer 1 that addresses this point (P4 L6-10):*

"Seedlings and saplings of both species are rare on Guam relative to nearby islands, consistent with the prediction that disperser loss reduces plant recruitment. However, the abundance of introduced ungulate herbivores is also highest on Guam, which makes it is difficult to attribute recruitment failure solely to mutualism disruption. Therefore, we designed our study to isolate the role of dispersers."

The authors should also frame their study in relation to defaunation. There are many studies on defaunation consequences on seed dynamics.

**While we recognize that there have been several strong studies focused on how defaunation affects seed rain and seedling recruitment, we decided to frame this study in terms of indirect effects of invasive predators. We have referenced several of the defaunation studies (P3 L18-19),*

including Harrison et al's study from defaunated forests in Borneo, a Cordeiro and Howe paper from Tanzania, and the McConkey et al review that summarizes impacts of defaunation on recruitment.

Details:

Fig 2 legend. Refer to e) and f).

**Done. Thanks for pointing out this mistake.*

Fig 3. Not clear why the position of trees shown as case studies is the same in Guam compared to the other islands. This is not at all possible.

**We simulated seed rain in a single plot based on dispersal kernels either from Guam or from islands with birds. By using the same adult tree distribution, we avoid complicating the figure with different adult tree distributions AND different seed rain patterns. We have amended the figure legend to better reflect this decision (P27 L 5-16).*

Reviewers' Comments:

Reviewer #1 (Remarks to the Author):

The authors have adequately addressed the issues raised in my review, and in my opinion the issues raised in the other two reviews. This remains an interesting and novel paper.

Reviewer #3 (Remarks to the Author):

I'm quite happy with the authors responses except that they need to downtone a bit the claimed crucial role of this indirect effect because other invasive species are also responsible for the decline of the species, and even if the dispersal birds could be "reintroduced", the plant species would still have fitness limitations. Referees asked for some observational data on seedling recruitment and they answered: "Seedlings and saplings of both species are rare on Guam relative to nearby islands, consistent with the prediction that disperser loss reduces plant recruitment. However, the abundance of introduced ungulate herbivores is also highest on Guam, which makes it difficult to attribute recruitment failure solely to mutualism disruption. Therefore, we designed our study to isolate the role of dispersers." Therefore, for example in the abstract: "We conservatively estimate that the brown treesnake has caused a 61-92% decline in seedling recruitment." need to be changed to "We conservatively estimate that the brown treesnake might have caused a 61-92% decline in seedling recruitment"

Response to Reviewers' Comments on NCOMMS-16-13124-T

We have responded to the editor's and reviewers' comment below, in italics, following the asterisk.

EDITOR'S COMMENTS:

Your manuscript entitled "Mutualism disruption by an invasive predator" has now been seen again by our referees, whose comments appear below. In light of their advice I am delighted to say that we are happy, in principle, to publish a suitably revised version in Nature Communications under a Creative Commons open access license. We therefore invite you to revise your paper one last time to address the remaining concerns of Reviewer 3. At the same time we ask that you edit your manuscript to comply with our format requirements and to maximise the accessibility and therefore the impact of your work.

REVIEWERS' COMMENTS:

Reviewer #1 (Remarks to the Author):

The authors have adequately addressed the issues raised in my review, and in my opinion the issues raised in the other two reviews. This remains an interesting and novel paper.

**Thank you*

Reviewer #3 (Remarks to the Author):

I'm quite happy with the authors responses except that they need to down tone a bit the claimed crucial role of this indirect effect because other invasive species are also responsible for the decline of the species, and even if the dispersal birds could be "reintroduced", the plant species would still have fitness limitations. Referees asked for some observational data on seedling recruitment and they answered: "Seedlings and saplings of both species are rare on Guam relative to nearby islands, consistent with the prediction that disperser loss reduces plant recruitment. However, the abundance of introduced ungulate herbivores is also highest on Guam, which makes it difficult to attribute recruitment failure solely to mutualism disruption. Therefore, we designed our study to isolate the role of dispersers." Therefore, for example in the abstract: "We conservatively estimate that the brown treesnake has caused a 61-92% decline in seedling recruitment." need to be changed to "We conservatively estimate that the brown treesnake might have caused a 61-92% decline in seedling recruitment".

** We have toned down our language as suggested in the abstract. In addition, we carefully read the remainder of the manuscript to look for similar instances, but did not find any other sentences that needed to be changed.*

.....